# DMPC Phospholipid Bilayer as a Potential Interface for Human Cystatin C Oligomerization: Analysis of Protein-Liposome Interactions Using NMR Spectroscopy

**DOI:** 10.3390/membranes11010013

**Published:** 2020-12-24

**Authors:** Przemyslaw Jurczak, Kosma Szutkowski, Slawomir Lach, Stefan Jurga, Paulina Czaplewska, Aneta Szymanska, Igor Zhukov

**Affiliations:** 1Faculty of Chemistry, University of Gdańsk, Wita Stwosza 63, 80-308 Gdańsk, Poland; przemyslaw.jurczak@ug.edu.pl (P.J.); slawomir.lach@ug.edu.pl (S.L.); aneta.szymanska@ug.edu.pl (A.S.); 2NanoBioMedical Centre, Adam Mickiewicz University, Wszechnicy Piastowskiej 3, 61-614 Poznań, Poland; kosma.szutkowski@outlook.com (K.S.); stjurga@amu.edu.pl (S.J.); 3Intercollegiate Faculty of Biotechnology UG & MUG, University of Gdańsk, Gdańsk, Abrahama 58, 80-307 Gdańsk, Poland; paulina.czaplewska@ug.edu.pl; 4Institute of Biochemistry and Biophysics, Polish Academy of Sciences, Adolfa Pawińskiego 5A, 02-106 Warszawa, Poland

**Keywords:** human cystatin C, NMR spectroscopy, ^15^N relaxation, phospholipid, DMPC, liposome, protein–liposome interaction

## Abstract

Studies revolving around mechanisms responsible for the development of amyloid-based diseases lay the foundations for the recognition of molecular targets of future to-be-developed treatments. However, the vast number of peptides and proteins known to be responsible for fibril formation, combined with their complexity and complexity of their interactions with various cellular components, renders this task extremely difficult and time-consuming. One of these proteins, human cystatin C (*h*CC), is a well-known and studied cysteine-protease inhibitor. While being a monomer in physiological conditions, under the necessary stimulus—usually a mutation, it tends to form fibrils, which later participate in the disease development. This process can potentially be regulated (in several ways) by many cellular components and it is being hypothesized that the cell membrane might play a key role in the oligomerization pathway. Studies involving cell membranes pose several difficulties; therefore, an alternative in the form of membrane mimetics is a very attractive solution. Here, we would like to present the first study on *h*CC oligomerization under the influence of phospholipid liposomes, acting as a membrane mimetic. The protein–mimetic interactions are studied utilizing circular dichroism, nuclear magnetic resonance, and size exclusion chromatography.

## 1. Introduction

Amyloid is a name used for the description of a specific protein (or peptide) that aggregated into its insoluble form featuring characteristic fiber-like shape [1]. The phenomenon of protein aggregation is an important issue because in many cases it leads to the loss of protein activity which may result in a disease state. The diseases correlated with amyloid formation are generally named amyloidoses. The process that leads to these diseases was not yet unambiguously described but it is known that the amyloid is formed as a result of protein oligomerization. Research indicates that biological membranes have a strong impact on the amyloidogenic proteins and their oligomerization process [2,3,4]. Up to date, two main possible pathways of membrane-induced oligomerization were presented. The first one assumes that a protein oligomerizes in the extracellular matrix, and the second indicates the membrane as the interface where the oligomerization occurs. Both the processes involve the formation of different oligomeric states of the protein, leading eventually to the formation of an insoluble fibril [5]. One of the oligomeric forms that are suspected to be the main cause of the amyloidoses is an annular oligomer that may interact with membranes forming channels which disturb the membrane integrity [5]. The formation of such structures has been observed for many amyloidogenic proteins and peptides like, among others, amyloid β peptide (Aβ) [6], immunoglobulin light chains [7], and human cystatin C (*h*CC) [8]. The last of the mentioned proteins is the subject of our interest. Up to date, it is not known if annular oligomers of *h*CC bind to the cell membrane and form channels. The process of *h*CC interaction with the membranes constitutes an interesting and important target of studies. It may explain the process of *h*CC toxicity against cells and its consequences, e.g., the processes leading to the occurrence of hereditary cystatin C amyloid angiopathy [9] or Alzheimer’s disease [10].

The studies of protein–membrane interactions are, however, complicated, time-, and cost-consuming experiments when performed on biological membranes isolated from natural sources. Previously, we performed a structural and dynamic analysis of the membrane mimetic—mixed micelle combining zwitterionic dodecylphosphocholine (DPC) and anionic sodium dodecyl sulfate (SDS) components [11]. In fact, the eukaryotic cell membranes are very heterogeneous. Various types of zwitterionic alkyl phosphatidylcholines with different alkyl chain lengths present in both membrane leaflets constitute a substantial part of the membrane lipid content [12,13]. What is important is that the distribution of lipids within the membranes is asymmetrical and controlled depending on their structure and function (e.g., promotion of blood coagulation) [14]. Such complexity of the membrane composition and control over lipid distribution is difficult to achieve in membrane mimetics. Therefore, compromises are being sought to simulate the membrane properties at least partially. In case of above-mentioned mixed micelle, the addition of small amount of negatively charged SDS provided better mimicry of electrostatic properties of the vertebrae cell membrane, which is characterized by slight prevalence of negative charge [11].

The number of techniques allowing to study interactions between proteins and membrane mimetics is rather scarce. Current techniques involve electrochemical techniques [15,16], microscopy techniques [17,18,19], light scattering techniques [20], etc. Among them, NMR spectroscopy is often utilized as it allows for spatial and structural analysis of (bio)molecules in various conditions, including biomembranes mimetics [21,22,23].

The studies performed so far revealed that in solution the wild type *h*CC protein (13 kDa, physiological pI = 9.3) reaches an equilibrium state between the monomer, being the major component, and the dimer. This results in two sets of overlapping signals observed in the NMR spectra, making them difficult for interpretation [24]. Similar to the wild type protein, performing the structural analysis for the *h*CC V57P variant (stable dimer) is a demanding task as the subunits of the dimeric variant are virtually identical, once again causing an overlap of NMR signals. As a result, the assignment of the backbone resonances was not yet performed for *h*CC V57P dimer. The oligomeric state of *h*CC is important in context of its structural studies since any interaction with ligands may cause changes in the oligomeric equilibrium of the amyloidogenic protein. However, due to technical difficulties (NMR signal overlap) at present, NMR measurements can be performed on stable monomeric *h*CC V57G variant only.

In this study, we used the *h*CC V57G mutant (stable monomer) to explore the structural and dynamic aspects of the interactions between the *h*CC V57G protein and the DMPC phospholipid bilayer. For the purpose of this study, DMPC was chosen as a membrane mimetic due to its high abundance in mammalian membranes and relatively easy procedure of liposome preparation [25]. DMPC is one of the commonly used mammalian membrane mimetics [26,27,28,29]. Even though it exhibits some limitations (e.g., shorter chain length than average lipids in human membranes), its properties to cost ratio presents it as a good candidate for initial studies on interactions between protein and membrane mimetics. We used size exclusion chromatography and circular dichroism to verify the protein stability in monomeric state and monitor any potential changes in its oligomeric state or secondary structure, which would indicate interactions with the DMPC. The NMR data allowed us to determine the amino acids in the *h*CC V57G sequence involved in interaction with DMPC bilayer.

## 2. Results and Discussion

### 2.1. Circular Dichroism

Circular dichroism (CD) spectroscopy experiments were performed to visualize the changes in the secondary structure of the *h*CC protein resulting from the interactions of the protein with the dimyristoylphosphocholine (DMPC) phospholipid. Recording of the CD spectra turned out to be difficult, due to high light scattering propensities of the DMPC liposome solution. However, after the prolonged process of liposome formation (twenty cycles of 30 min incubation in ultrasound bath with heating and 30 min incubation in 4 ${}^{{}^{\circ}}$C), followed by extrusion, it was possible to obtain a solution transparent enough to register CD spectra for the *h*CC V57G protein in the presence of DMPC at various concentrations (Figure 1). Collected spectra revealed the stability of the *h*CC V57G protein as no changes in its structure were observed, regardless of the temperature of incubation (22 ${}^{{}^{\circ}}$C or 37 ${}^{{}^{\circ}}$C) and DMPC concentration. The stability of *h*CC V57G agrees with our previous data showing that the protein is stable even at increased temperatures and at low pH values, which cause conformational changes and dimerization of the native *h*CC protein [24]. Contrary to the data obtained for interaction between *h*CC V57G and micellar membrane mimetics (DPC and SDS), no increase of α-helical secondary structure was observed within the protein (data accepted by *Membranes*). It is possible that the interactions between *h*CC V57G and the liposomes are rather weak and require higher liposome concentrations for the changes of protein structure to be observed. On the other hand, performing the experiments at higher DMPC concentrations proved not to be possible due to the high light-scattering properties of the solution.

Most of the phospholipid-based liposomal solutions (e.g., DMPC, POPC, etc.) have a tendency for strong light scattering. The only way to make the solutions more transparent is to decrease the liposome size, as the intensity of light scattered by a sphere is directly related to its radius [30]. In experimental conditions, this was achieved by increasing the temperature up to 40–50 ${}^{{}^{\circ}}$C during sonication (see Section 4.2). However, what was unexpected in the case of CD experiments was that decreasing the temperature of the solution after liposome preparation caused the increase of light scattering again and hindered data collection. Performing the experiments at 40–50 ${}^{{}^{\circ}}$C would be, however, physiologically irrelevant; therefore, we did not attempt these studies at such temperatures.

### 2.2. Size Exclusion Chromatography

Size exclusion chromatography (SEC) experiments were performed to verify if there are any changes in the oligomeric state of *h*CC protein caused by the presence of different concentrations of DMPC liposomes, indicating interactions between the molecules and possible impact on further oligomerization of protein. The incubation of the *h*CC V57G with DMPC solutions did not cause any particular changes in the oligomeric state of the analyzed protein. No changes in the intensity of signals or retention times were observed in the chromatograms indicating that, regardless of the incubation temperature, the *h*CC V57G monomer is stable in the presence of DMPC liposomes (Figure 2A,B). This further confirms the high stability of *h*CC V57G mutant in different environmental conditions in contrast to other *h*CC variants which may dimerize or oligomerize depending on the environmental factors [24,31]. The *h*CC–liposome interactions observed with NMR spectroscopy would indicate that during size exclusion chromatography the *h*CC V57G bound to the liposome should elute from the SEC column somewhere around the elution time characteristic for the liposomes—*h*CC is ~30 times smaller than the liposome, so the elution time for the complex should not significantly differ from the liposomes’ elution time. This would result in the decrease of the *h*CC signal in the chromatogram. Such phenomenon was however not observed. We assume that the interaction observed with NMR is rater weak and a fast exchange between bound and unbound states occurs. Therefore, when the solution containing lipid and *h*CC flows through the size exclusion chromatography column the components are being physically separated from one another when in unbound state.

### 2.3. NMR Spectroscopy and Assignment of the ${}^{1}$H and ${}^{15}$N Backbone Resonances

The structural and dynamic aspects of the interactions between *h*CC V57G and cell membrane were monitored by NMR spectroscopy using the DMPC phospholipid as a membrane model. The ${}^{1}$H-${}^{15}$N HSQC spectrum acquired in the presence of DMPC-$d_{54}$ demonstrates that signals coming from amide groups of the *h*CC V57G exhibit good dispersion, confirming that in this experimental condition the *h*CC V57G is properly folded (Figure 3). As expected, the assignments of the majority of ${}^{1}$H and ${}^{15}$N resonances were obtained based on previously acquired data (bmrb 34399) [24]. For several amide groups, the degeneration of ${}^{1}$H or ${}^{15}$N chemical shifts was resolved after acquiring the 3D ${}^{15}$N-edited TOCSY-HSQC and ${}^{15}$N-edited NOESY-HSQC experiments. Finally, our attempts resulted in assignments of the ${}^{1}$H and ${}^{15}$N backbone resonances for the 97 amide groups out of 111 expected. Inspection of the evaluated assignments, and comparing them with the data reported previously reported for the *h*CC V57G in solution, demonstrate the presence of the DMPC-$d_{54}$ phospholipid lead only to tiny alterations of the 3D protein structure.

The analysis of the *h*CC V57G mutant by PGSE-NMR deliver an additional confirmation of the stability of the 3D structure at different experimental conditions. The translation diffusion coefficient ($D_{tr}$) extracted from PGSE experiment equals $D_{tr}$ = 1.14 ± 0.03 × 10${}^{-10}$ (m${}^{2}$/s) (Appendix A). The obtained value is close to the $D_{tr}$ reported for several variants of *h*CC protein in solution [32].

### 2.4. Backbone Dynamic of hCC V57G Based on the ${}^{15}$N Relaxation Data

The analysis of ${}^{15}$N relaxation data was performed with a model-free approach combined with axially anisotropic overall molecular tumbling [33,34]. Four global parameters—$D_{∥}$, $D_{⊥}$, θ, and ϕ—describe parallel and perpendicular components of the rotational diffusion tensor and direction of the unique axis of the rotation diffusion tensor of the biomolecule, respectively. Three local, residue-specific parameters comprise a generalized order parameter *S*, which is a measure of the degree of spatial restriction of the motion, an effective correlation time $\tau_{int}$ corresponding to the rate of this motion, and $R_{ex}$ describing conformational exchange contribution to $R_{2}$ resulting from the dynamic processes on a microsecond to millisecond time scale [35].

The experimental ${}^{15}$N relaxation data were used to extract $R_{1}$, $R_{2}$, and ${}^{1}$H-${}^{15}$N NOE for the 82 resonances (Figure 4). The relaxation parameters were not calculated for several signals which are substantially broadened or even not observed at the ${}^{1}$H-${}^{15}$N HSQC spectrum due to fast exchange with water.

The structural analysis performed for *h*CC V57G in solution reveals that rotational diffusion tensor can be presented as prolate axially symmetric [24]. The $D_{∥}$ and $D_{⊥}$ main components of the rotational diffusion tensor can be calculated from the ratio of measured $R_{2}/R_{1}$ relaxation rates [36,37]. The $R_{2}/R_{1}$ data obtained from experimental data resulted in $D_{∥}$= 2.63 ± 0.02 × 10${}^{7}$ (s${}^{-1}$) and $D_{⊥}$= 2.06 ± 0.02 × 10${}^{7}$(s${}^{-1}$) as main components of the axially symmetrical rotational diffusion tensor. The rotation correlation time was calculated as $\tau_{R}=1/\left(2D_{∥}+4D_{⊥}\right)$ and equaled 7.41 ± 0.06 ns which fits well with the value for 120 residue long protein, and is close to previously reported $\tau_{R}$ expected for the *h*CC V57G in solution [24].

The secondary structure of *h*CC V57G comprises two α-helices (Glu21–Tyr34, and Asn82–His90) and five β-strands (Met14–Asp15, Gln48–Ile56, Val60–Leu68, Ala95–Val104, and Thr109–Thr116). The residues form the elements of secondary structure characterized by NOEs higher than 0.8 (Figure 4). At the same time, residues in flexible loops (Gly57–Gly59 and Pro105–Gly108) demonstrate the NOE reflecting higher flexibility for those structural regions.

The values of local parameters—$S^{2}$ and $R_{ex}$—together with $R_{1}R_{2}$ product are presented in Figure 5. The higher values for $S^{2}$ were observed for the residues located in α-helices (green bars, $S_{av}^{2}$ = 0.90), and β-sheet (blue bars, $S_{av}^{2}$ = 0.91). The flexibility of the N-terminal part of the protein is clearly indicated with lower values of $S^{2}$ parameter (red bars, $S_{av}^{2}$ = 0.77). Interestingly, the Gly57 (mutation site) located in the hairpin structure between two anti-parallel β-strands exhibits the $S^{2}$ value of 0.57—substantially lower compared to the neighboring residues (Figure 5).

The transverse relaxation rates ($R_{2}$) included the slow dynamic processes ($R_{ex}$) observed in low-frequency—milli- or microsecond time-frame (10${}^{-3}$–10${}^{-6}$ s). Although the $R_{ex}$ term was detected for residues located along whole *h*CC V57G sequence, we can conclude the low intensity of chemical exchange (taking into account that $R_{ex}$ scales with square of magnetic field) with the exception of Asp119 in the C-terminus (Figure 5).

An additional possibility of fast and simple identification of the residues undergoing conformational exchange slower than the overall molecular tumbling can be derived from analysis of $R_{1}R_{2}$ product [38]. The $R_{1}R_{2}$ data is more reliable compared to the $R_{1}/R_{2}$ ratio, which is prone to errors in the case of anisotropic overall tumbling. During the inspection of collected data, we have selected the residues exhibiting larger than average $R_{1}R_{2}$ values as residues with increased mobility in low-frequency time frame (Figure 5). Inspection of the $R_{1}R_{2}$ relation dependence allowed us to select three fragments with such motions—Ser44, Asn82, and Asp119—which are located in the same structural region. The possibility to collect NMR data for *h*CC, which should tumble rather slow after binding to the relatively large liposome, can be explained by relatively weaker interactions between the *h*CC V57G and the DMPC liposomes. The exchange rate between bound and free states is very fast (probably much faster compared to NMR observation time-fast regime). As a result, only a small amount of molecules in the bound state are present in the solution. Therefore, the rotational and translational motions observed during the measurement are very similar to those observed for the V57G variant in solution. The relaxation parameters, on the other hand, are quite sensitive to the existence of a transition state, even a couple of percent of a different conformation can be detected by the analysis of relaxation data.

### 2.5. hCC V57G Interactions with DMPC Liposome

The experimental data acquired for *h*CC V57G in the environment of DMPC-$d_{54}$, compared with the data from previously performed NMR backbone sequential assignment for the same protein in solution (pdb 6RPV) [24], suggest that the presence of the DMPC leads to rather limited structural alterations of the protein. Detailed analysis of the experimental data showed that the presence of phospholipids in the solution lead to local changes in csp (Figure 6A,B), which clearly highlighted the structural regions which interacted with the DMPC liposome (Figure 6C). The majority of amide groups which showed the highest csp values were located in the N-terminus in close proximity to the beginning of α-helical part of the protein–${}^{16}$ASVEEEGV${}^{23}$ and β-strand motif (${}^{51}$RARKQ${}^{55}$), which are both located on the outside of the protein and easily accessible form solution.

The backbone mobility in a low-frequency time frame is another important aspect of *h*CC V57G interaction with the DMPC liposome. The presence of the DPMC-$d_{54}$ in solution was analyzed on the base of the $R_{1}R_{2}$ product (Figure 7A). Inspection of the presented data showed several residues with increased structural exchange dynamic processes—Gly69, Lys75, and Ala95. Those three residues combined with previously pointed residues exhibited higher values of $R_{ex}$ term on $R_{1}R_{2}$ relation dependence were presented in Figure 7B. All highlighted residues were located in the partially disordered AS region (appending structure) connecting β3 and β4-strands [24]. They provide additional possibility of regulation of *h*CC interactions with the DMPC phospholipid.

## 3. Conclusions

The presented study focused on the description of the interaction sites between *h*CC V57G and DMPC membrane mimetic. The acquired experimental data demonstrated that the *h*CC V57G variant does not migrate into the DMPC bilayer but locates itself on the lipid surface. The bilayer exhibits a rather weak attraction to the N-terminal part of the protein and β-sheet strand, both of which are located on the exterior of the protein, with easy access for the interaction. Concluding from the occurring interaction, the hypothesis placing biological membranes in the role of an interface inducing *h*CC oligomerization seems to be rational. However, to fully verify it further, detailed studies involving advanced membrane constructs, natural membranes, and *h*CC variants prone to oligomerization processes are required.

## 4. Materials and Methods

### 4.1. Expression and Purification of Labeled and Unlabeled Proteins

The DNA of *h*CC V57G variant was obtained using site-directed mutagenesis as previously described [31]. Plasmid DNA (pHD313 vector [39]) containing human cystatin C gene coupled with signal peptide derived from *E. coli* OmpA protein (causes the secretion of a protein into periplasmic space), temperature-sensitive λ cI 857 repressor, λ PR promoter, and ampicillin resistance gene was transformed to and expressed in *E. coli* BL21(DE3)-competent cells (Novagen) using the standard LB medium and temperature-induced expression, according to the protocol described earlier [31].

### 4.2. Liposome Sample Preparation

The lyophilized DMPC powder was suspended in PBS buffer (Sigma Aldrich, St. Louis, MO, USA). If the solution was used in NMR experiments, the buffer contained 10% D${}_{2}$O. The phospholipid suspension was subjected to 10 incubation cycles involving 30 min incubation in ultrasound bath with heating (313 K) and 30 min incubation in 277 K. Next, the extrusion with Avanti Mini Extruder (Avanti Polar Lipids, USA) with 0.1 μm pore membrane was performed in case of SEC and CD samples only, to decrease the probability of column clogging by relatively large lipid rafts in SEC or high light scattering in CD experiments. This procedure, i.e., passing the lipid solution through the extruder, was repeated eleven times. It should be noted that the solution should always be forced through the extruder an odd number of times [11]. After the extrusion was finished, the sample was ready for use. For the purpose of the experiments, 10 mM molar DMPC liposome stock solution was prepared and used for preparation of a dilution series.

### 4.3. Circular Dichroism Spectroscopy

The *h*CC V57G protein was dissolved at the concentration of 10 μM in PBS buffer. It was next incubated for 24 h at 295 K (room temperature) and 310 K (human body temperature) with DMPC at different phospholipid concentrations (1:1 *v*/*v* five step dilution series with the highest concentration of $C_{DMPC}$ = 5 mM; liposome size 100 nM—size of the pores in the extruder filters). Before the experiment, the samples were spun at 10,000× *g* for 5 min to remove any insoluble particles from the solution. The CD spectra were registered with JASCO J-815 spectropolarimeter for supernatants at 295 K in the UV range of 195 to 260 nm and analyzed with Jasco Spectra Manager software.

### 4.4. Analytical Size Exclusion Chromatography

The *h*CC V57G protein was dissolved at the concentration of 30 μM (in PBS) and incubated with DMPC at different phospholipid concentrations (1:1 five step dilution series with the highest concentration of $C_{DMPC}$ = 5 mM), for 24 h and at different temperatures (295 K—room temperature, 310 K—human body temperature). After incubation 10 μL of the mixture was applied on the gel filtration column (Superdex 75 PC 3.2/30, GE Healthcare Life Sciences) and eluted with PBS buffer (flowrate 0.1 mL/min).

The results were analyzed with Chromax 2007 (POL-LAB, Poland) and OriginPro 2018 (OriginLab Corporation, Northampton, MA, USA) software to verify if the *h*CC monomer–dimer equilibrium has changed.

### 4.5. NMR Sspectroscopy

NMR measurements were performed on Agilent DDR2 800 NMR spectrometer operated at magnetic field strength of 18.8 T (${}^{1}$H resonance frequency 799.786 MHz) installed in NanoBioMedical Centre at Adam Mickiewicz University (Poznań, Poland) equipped with three channels, *z*-gradient unit and ${}^{1}$H/${}^{13}$C/${}^{15}$N triple-resonances probe-head with inverse detection. All experimental data were recorded at 298 K processed with the NMRPipe [40] and analyzed with the Sparky [41] software. The ${}^{1}$H and ${}^{15}$N resonances were referenced to external sodium 2,2-dimethyl-2-silapentane-5-sulfonate (DSS) using coefficient Ξ = 0.101329118 for ${}^{15}$N dimension [42].

### 4.6. ${}^{15}$N Relaxation Measurements

The molecular dynamic processes in *h*CC V57G variant were analyzed with ${}^{15}$N relaxation experiments performed at the magnetic field of 18.8 T. The ${}^{15}$N longitudinal ($R_{1}$) and transverse ($R_{2}$) relaxation rates together with ${}^{1}$H-${}^{15}$N NOE values were collected with pulse sequences included in BioPack (Agilent, PaloAlto, CA, USA) written based on the previously published experiments [43]. The ${}^{15}$N $R_{1}$ data sets were acquired with eleven delays (10, 90, 170, 290 (twice), 410, 550, 690, 850, 1010, and 1250 ms). The ${}^{15}$N $R_{2}$ relaxation rates were obtained with nine delays (10, 30, 50, 70, 90 (twice), 110, 150, and 190 ms). The cross-correlation effect was suppressed by the delay between π(${}^{1}$H) pulses of 5 and 10 ms for $R_{1}$ and $R_{2}$ measurements, respectively [44]. The recycle delay for $R_{1}$ and $R_{2}$ experiments was establish as long as 3.5 s. The ${}^{15}$N $R_{1}$ and $R_{2}$ relaxation rates were calculated with a two-parameter model of a single exponent decay with the RELAX software [45] (version 4.0.1). The ${}^{1}$H-${}^{15}$N NOE values were obtained as the ratio of peak intensities in two 2D experiments performed with and without saturation. In the ${}^{1}$H-${}^{15}$N NOE measurements, the 6 s delay was used either as recycling or saturation ${}^{1}$H magnetization.

### 4.7. Analysis of ${}^{15}$N Relaxation Data with a Model-Free Approach

The nuclear spin relaxation of amide nitrogens in proteins is defined by two mechanisms—the dipole–dipole interaction between a ${}^{15}$N and the directly bound ${}^{1}$H, and the chemical shift anisotropy of a ${}^{15}$N nucleus. The experimental data, which are usually acquired for the analysis of ${}^{15}$N relaxation included the longitudinal ($R_{1}$), and transversal ($R_{2}$) relaxation rates, together with ${}^{1}$H-${}^{15}$N NOE. Equations describing measured relaxation parameters in terms of spectral density functions [34]: $$\begin{array}{c} R_{1}=\frac{1}{4}D^{2}\left[J\left(\omega_{H}-\omega_{N}\right)+6J\left(\omega_{H}+\omega_{N}\right)\right]+\frac{1}{3}C^{2}J\left(\omega_{N}\right)\\ R_{2}=\frac{1}{8}D^{2}\left[4J\left(0\right)+J\left(\omega_{H}-\omega_{N}\right)+3J\left(\omega_{N}\right)+6J\left(\omega_{H}+\omega_{N}\right)\right]+\frac{1}{18}C^{2}\left[4J\left(0\right)+3J\left(\omega_{N}\right)\right]+R_{ex}\\ NOE=1+\left(\frac{\gamma_{H}}{\gamma_{N}}\right)\frac{D^{2}}{4R_{1}}\left[6J\left(\omega_{H}+\omega_{N}\right)-J\left(\omega_{H}-\omega_{N}\right)\right] \end{array}$$
where J(ω) is the spectral density of molecular motion at a given angular frequency, $D=\frac{\mu_{0}}{8\pi^{2}}\frac{\gamma_{H}\gamma_{N}h}{\langle r_{NH}^{3}\rangle }$ is the strength of the dipolar coupling, $C=\frac{\omega_{N}}{\sqrt{3}}\Delta \sigma$ is the ${}^{15}$N CSA term, $\mu_{0}$ is the permeability of free space, *h* is Plank’s constant, $\gamma_{H}$ and $\gamma_{N}$ are the proton and nitrogen gyromagnetic ratios, respectively. The $r_{NH}$ is the amide bond length ($r_{NH}$ = 1.02 Å), and Δσ is the difference between parallel and perpendicular components of the assumed axially symmetric ${}^{15}$N chemical shift tensor (Δσ = −160 ppm).

An additional term $R_{ex}$, which scales with the square of magnetic field, reveals contribution to transverse relaxation rates ($R_{2}$) from processes in micro- to millisecond time scale, usually named conformational exchange [35]. Such processes, slower than the molecular tumbling, but fast enough to average out chemical shifts, can influence the $R_{2}$. The $R_{ex}$ contribution to the transverse relaxation rate is proportional to the square of the chemical shift difference between exchanging states, $R_{ex}\;=\;\Phi \omega_{N}^{2}$, where the factor Φ characterizes the effectiveness of conformational exchange processes [46].

In the case of isotropic molecular tumbling, the model-free approach spectral density function takes the form of [33]
$$J\left(\omega \right)=\frac{2}{5}\left[\frac{S^{2}\tau_{R}}{1+{\left(\omega \tau_{R}\right)}^{2}}+\frac{(1-S^{2})\tau }{1+{\left(\omega \tau \right)}^{2}}\right]$$
with $\tau^{-1}\;=\;\tau_{R}^{-1}+\tau_{c}^{-1}$. The isotropic overall motion is described by the correlation time $\tau_{R}$ and internal motion(s) by a generalized order parameter *S*, which is a measure of the degree of spatial restriction of the motion. An effective correlation time $\tau_{c}$ corresponding to the rate of this motion. In the case of anisotropic overall motion the spectral density function becomes more complex. Combining the model-free approach with axially anisotropic overall tumbling the spectral density function takes the form of [47]
$$J\left(\omega \right)=S^{2}\left[\frac{A_{1}\tau_{1}}{\left[1+{\left(\omega \tau_{1}\right)}^{2}\right]}+\frac{A_{2}\tau_{2}}{\left[1+{\left(\omega \tau_{2}\right)}^{2}\right]}+\frac{A_{3}\tau_{3}}{\left[1+{\left(\omega \tau_{3}\right)}^{2}\right]}\right]+\frac{(1-S^{2})\tau }{\left[1+{\left(\omega \tau \right)}^{2}\right]}$$
where α is the angle between N-H vector and the unique axis of rotational diffusion tensor. The overall correlation times ($\tau_{k}$) are defined as $\tau_{1}=1/\left(4D_{∥}+2D_{⊥}\right)$, $\tau_{2}=1/\left(D_{∥}+5D_{⊥}\right)$, and $\tau_{3}=\left(6D_{⊥}\right)$. $D_{∥}$ and $D_{⊥}$ are parallel and perpendicular components of the rotational diffusion tensor.

### 4.8. Translational Diffusion Measurement with PGSE-NMR

Diffusion experiments were conducted on an Agilent DDR2 800 NMR spectrometer utilizing a standard PGSE (Pulsed Gradient Spin Echo) pulse sequence [48] containing a solvent suppression block [49]. The measurements were performed using diffusion time (Δ) of 150 ms. Translation diffusion coefficient ($D_{tr}$) was calculated using 25 gradients with effective gradient pulse duration (δ) 3 ms. The 96 accumulations were performed with a recycling delay 1.8 s in order to increase the signal-to-noise ratio. The amplitude signals observed between 0 and 3 ppm were used for extraction of the $D_{tr}$ value. The $D_{tr}$ was calculated by fitting using Stejskal–Tanner equation taking into account an additional correction for the Δ delay during BPP pulse in sequence (delay between gradient δ and π/2 pulse was equal to 0.5 ms) [50]:$$I\;=\;I_{0}\exp\left(-D{\left(G\gamma \delta \right)}^{2}\left(\Delta -\delta /3\right)\right)$$
where $\gamma_{H}$ is the ${}^{1}$H gyromagnetic ratio, δ is the gradient duration, Δ is the diffusion time, and *G* is the gradient strength. The $D_{tr}$ was obtained from (Appendix A).

## Figures and Tables

**Figure 1 membranes-11-00013-f001:**
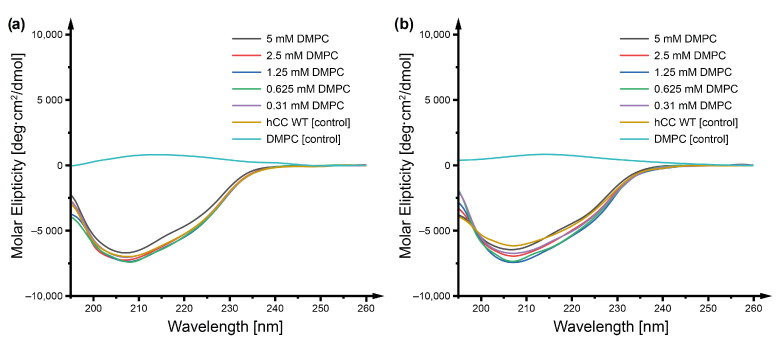
Circular dichroism spectra visualizing changes of the secondary structure of *h*CC V57G influenced by the environment DMPC liposome solution after 24 h of incubation at (**A**) 22 ${}^{{}^{\circ}}$C and (**B**) 37 ${}^{{}^{\circ}}$C. The *h*CC:liposome ratio calculated for $C_{DMPC}$ = 5 mM, equaled 16 *h*CC molecules per liposome.

**Figure 2 membranes-11-00013-f002:**
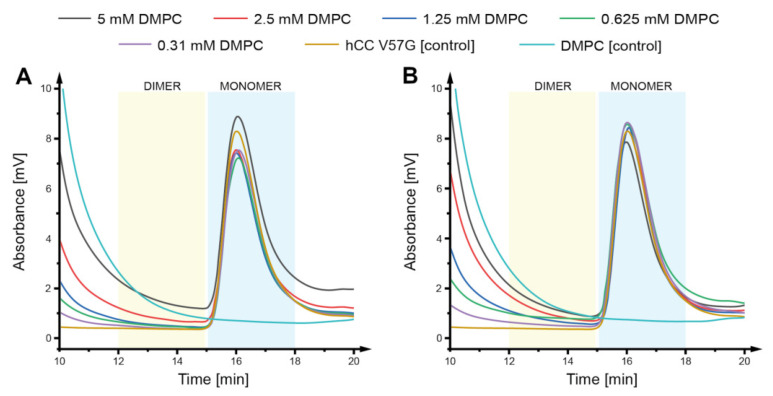
Chromatograms visualizing separation of *h*CC V57G monomer on the SEC column after incubation at (**A**) 22 ${}^{{}^{\circ}}$C and (**B**) 37 ${}^{{}^{\circ}}$C for 24 h in the DMPC solution; monomer retention time—16 min. The *h*CC:liposome ratio calculated for $C_{DMPC}$ = 5 mM, equaled 48 hCC molecules per liposome.

**Figure 3 membranes-11-00013-f003:**
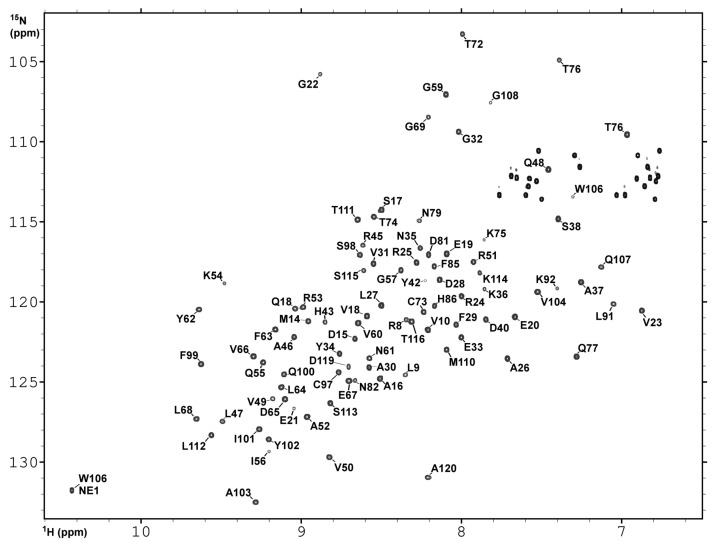
The ${}^{1}$H-${}^{15}$N HSQC spectrum acquired for *h*CC V57G in the presence of DMPC-d54 on Agilent DDR2 NMR spectrometer at 298 K. The assignments of ${}^{1}$H and ${}^{15}$N backbone resonances are presented as one-letter code and sequence number.

**Figure 4 membranes-11-00013-f004:**
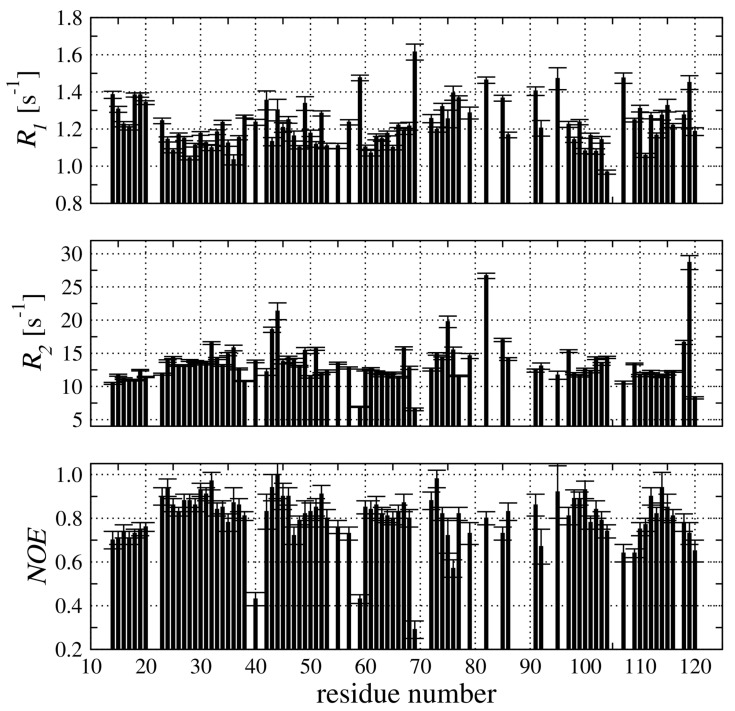
The $R_{1}$, and $R_{2}$ relaxation rates together with ${}^{1}$H-${}^{15}$N NOE experimental data acquired at 18.8 T and 298 K, utilizing an Agilent DDR2 NMR spectrometer.

**Figure 5 membranes-11-00013-f005:**
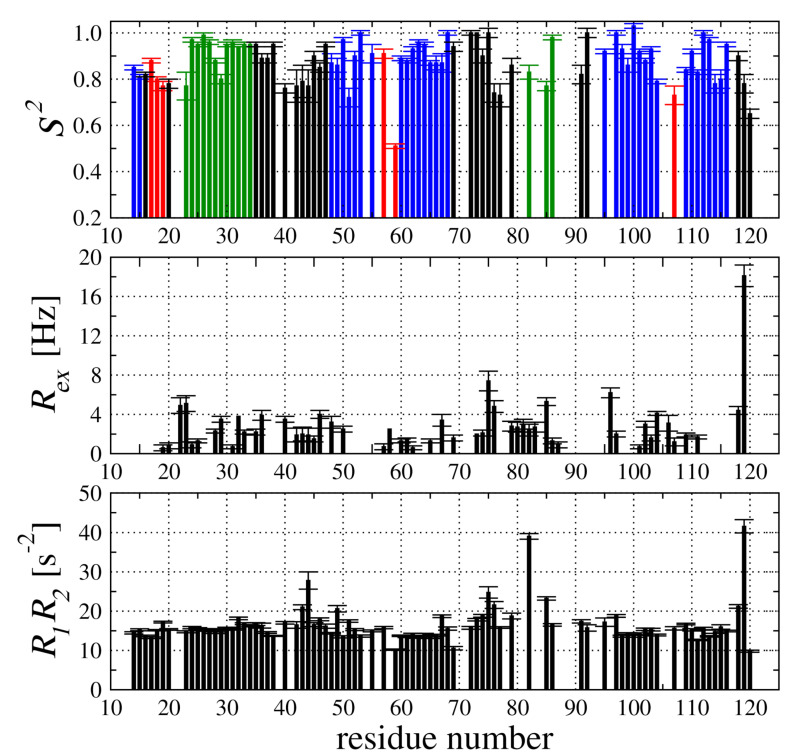
Model-free approach parameters: $S^{2}$ (top), $R_{ex}$ (middle), and $R_{1}R_{2}$ product (bottom). The $S^{2}$ parameter is highlighted for the residues in different secondary structural elements as green (α-helix), blue (β-sheet), and red (flixible loops).

**Figure 6 membranes-11-00013-f006:**
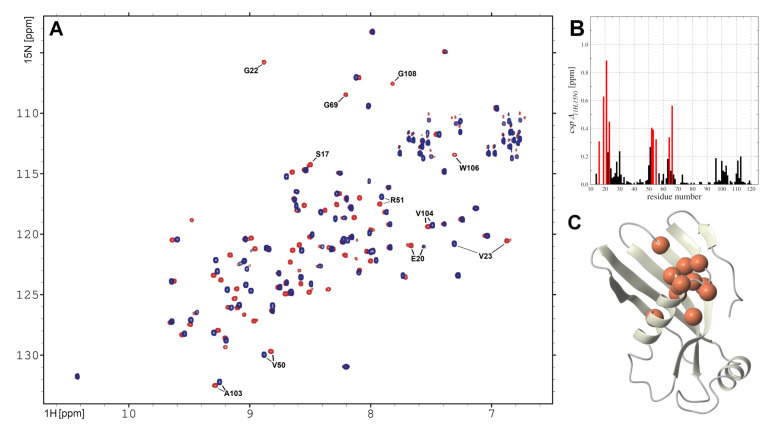
(**A**) The overlay of ${}^{1}$H-${}^{15}$N HSQC spectra acquired for *h*CC V57G variant in solution without (blue) and with DMPC-$d_{54}$ (red); (**B**) The csp plot for ${}^{1}$H and ${}^{15}$N chemical shifts in sequence-specific manner, with amino acid residues taking part in interaction with DMPC-$d_{54}$ (csp > 0.2) marked red; (**C**) Model of *h*CC V57G protein; amino acids exhibiting csp > 0.2 marked as red marbles.

**Figure 7 membranes-11-00013-f007:**
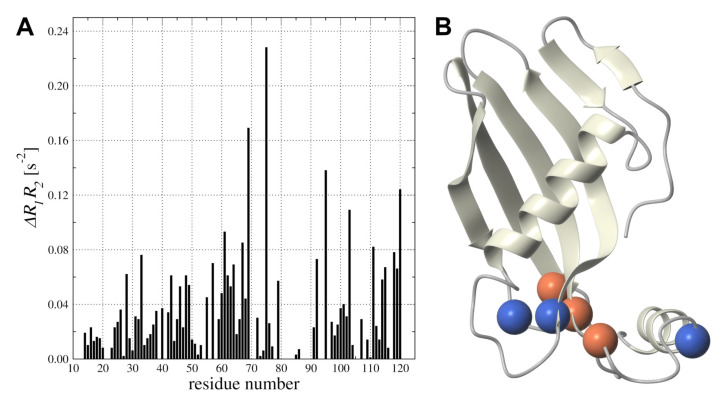
(**A)** Differences in $R_{1}R_{2}$ product calculated on the base of $R_{1}$ and $R_{2}$ relaxation rates measured in solution before and after addition of DMPC-$d_{54}$; (**B**) Ribbon representation of the 3D structure of *h*CC V57G with residues exhibiting increased conformation exchange mobility (Ser44, Asn82, and Asp119) and changes after addition DMPC-$d_{54}$ (Gly69, Lys75, and Ala95) shown as blue and red marbles, respectively.

## Data Availability

Data is contained within the article or supplementary material.

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
