# Peer review of "DMPC Phospholipid Bilayer as a Potential Interface for Human Cystatin C Oligomerization: Analysis of Protein-Liposome Interactions Using NMR Spectroscopy"

_membranes, 2020, doi:10.3390/membranes11010013_

Round 1

Reviewer 1 Report

The paper itself presents an interesting study on the interactions of human cystatin with a model for the cellular membrane (DMPC) using 1HNMR and 15NNMR. It is certainly of interest to the membrane biophysics community to see more of these investigations, particularly as extensive experience and knowledge in this field is required to be able to carry out these experiments.

The visual representations of the interaction sites in Figure 6 and 7 were very valuable to illustrate the findings of the experiments.

The introduction does not mention some important work in this area, for example a highly cited paper by Valincius et al (https://doi.org/10.1529/biophysj.108.130997), which should especially be mentioned when they discuss electrochemical techniques in line 46 or the review by Sakono et al (https://doi.org/10.1111/j.1742-4658.2010.07568.x).

It was not clear to me, why DMPC was chosen over other phospholipids. This should perhaps be justified in the introduction, particularly as phosphocholines and cholesterol are also present in the membranes of brain cells and may affect interactions.

It was also not clear, if the NMR experiments were performed with dissolved lipid or lipid vesicles. As the real interaction presumably involves the cell membrane rather than dissolved lipids, a study examining the interaction between cystatin and free lipid would be of limited physiological relevance.

In the methods section, liposome preparation using only 10 extrusion steps was described - this is at the lower limit of what is recommended by the manufacturer of the extruder. Typically, extrusion is repeated 20 times or more to obtain closer to monodisperse solutions. The authors should clarify if they needed monodisperse solutions for these experiments. Either way, vesicle size should be confirmed using DLS before and after the NMR experiments to ensure they were stable.

In the circular dichroism section, no statement is made regarding vesicle size. This should be added, particularly as small vesicles (the authors describe  using 100 nm filters) are, in terms of membrane curvature, not reflective of real cell membranes. If vesicles were used for the NMR experiment, vesicle diameter should also be shown and it should be discussed why the diameter in question (100 nm?) was chosen rather than a more biologically relevant diameter which would also increase vesicle stability.

In this section, the authors also refer to as yet unpublished data in this section. I would suggest waiting until the other publication has been accepted for publication before accepting this paper or at the very least updating the statement in line 73 with a proper reference as soon as possible after publication.

To represent the data in Fig. 7A, more contrasting colours than dark blue and dark red should be chosen to make the graph easier to read.

Reviewer 2 Report

Minor points:

Line 4 – repetition of “complexity”?

Line 42 – “phospholine” – do you me “phospatidylcholine”?

Lines 42-44 – there needs to be more narrative around the asymmetric nature of the membrane and the relative depletion of PC on the inner membrane.  See, for example, https://dx.doi.org/10.1080%2F10409230903193307 and papers that cite this.  I am not clear how addition of SDS helps “better simulate” a membrane?

Line 176 – “sites” not “sights”.

I think the whole last paragraph of the introduction needs rewriting – it should state exactly what is to be done in this study.  I think the information about the stable dimer needs more explanation.  It may be worthwhile giving a very brief summary of the key findings of the paper here.  I think it needs to be much clearer what is being done and why.  This particularly pertains to the use of the mutant protein throughout these experiments.

Major comments:

Line 62 – there needs to be a clear rationale for the choice of DMPC as a “mimetic” of a eukaryotic membrane.  It is actually a poor mimetic both as the head groups in eukaryotic cells are much more diverse than just PC.  Additionally, the chain lengths in DMPC are shorter than the average in e.g. humans (16-18C).  This needs to be addressed and the limitations recognised.

Line 65 – do the authors know what the average size of their liposomes is?  I suspect, given that they have sonicated extensively to avoid scattering at the wavelengths used for CD, the liposomes will be <<100nm in diameter.  This is an issue as they are highly curved and not representative of eukaryotic membranes (several orders of magnitude difference in diameter).

Line 68 – the temperatures here are critical.  The phase transition temperature of DMPC is 24oC so it will be in different phases in the two experimental setups and therefore results are not comparable.  This should be explicitly commented on.

Figure 1 – it would be good to include the protein:lipid ratio here to give an idea of the number of proteins per lipsosome.

Line 78-79 – this is completely incorrect, the best way to reduce scattering is to decrease liposome size through e.g. controlled extrusion.  Changing the lipid phase has minimal impact on scattering as that is a physical property of the particle.

Figure 2 – it seems like the liposomes are likely coming through in the void volume of the column – if the protein really associates with the liposomes then it would come out with them there.  Is there any loss of protein from the monomer peak that could be investigated in the void volume?  This should be discussed.

Figure 4 – the temperature chosen is really close to the phase transition temperature of DMPC – this is far from ideal for this sort of experiment and needs to be discussed.

Figure 6 – I am surprised that proteins that contact the liposomes (and are attached to them) tumble quickly enough to be seen in an NMR experiment.  This should be commented on.

Nevertheless, I think it is important to try to rationalise why the residues identified may be involved in lipid interactions.  I think this section needs more interpretation.

Methods – it is important to size the liposomes by e.g. DLS after extrusion.  If they are too small to start with e.g. 30-50nm they will undergo rapid fusion.

I am not convinced by the CD or SEC data for the reasons outlined above.  The NMR data looks ok but needs much more rationalisation to inform future studies.  I think the limitations of the lipid identity and the temperatures chosen need to be considered throughout.

Reviewer 3 Report

In their original research article, Przemyslaw Jurczak and colleagues explore, using molecular circular dichroism, size exclusion chromatography and NMR, the lipid-protein interactions between DMPC bilayers and human cystatin C V57G. Even though it seems to be weak, the Authors show that the interaction is restricted to the bilayer surface, mostly be means of NMR data. The data presented does not exclude the hypothesis that a phosphatidylcholine lipid bilayer can act as an interface catalysing the oligomerization of hCC. Prior to publishing however, the Authors should address several concerns in order to improve the readability, impact and consistency of their work.

Broad comments

1)     Whenever is missing the Authors must clarify in each Figure subtitle if the data is obtained in the presence of a DMPC bilayer or not. For example, it should be mentioned in the legend of Figure 3 that the data was collected in a solution containing DMPC bilayers.

2)     Some of the physicochemical properties of the hCC protein used in this work should be mentioned. For example, its size, in terms of kDa, or its isoelectric point, if already determined.

3)     Regarding the circular dichroism experiments, the Authors mention that performing experiments at high DMPC concentrations was not possible due to light scattering.  Perhaps it is possible to overcome this limitation by adopting an opposite strategy, i.e., decrease the amount of protein until the same protein:lipid proportions are achieved. However, one possible disadvantage of this approach might be a decrease in the CD signal.

4)     Some English correction is necessary throughout the manuscript. Few examples are: line 129 “as a main components” – “as main components”; line 130 “equaled 7.41_0.06 ns which fit well” – “equaled 7.41_0.06 ns which fits well”; line 133 “presented as two α-helices” – “presented two α-helices”; line 139-140: “The higher value for S2 observed for the residues located in…” – “The higher values for S2 were observed for the residues located in…”; line 188: “causes the secretion of into periplasmic…” – “causes the secretion into periplasmic” or “causes the secretion of hCC into periplasmic…”

5)     Although the wild type protein presents a fast-forming equilibrium between the monomer and the dimer, which generates two sets of overlapping signals in the NMR spectra, it would have been interesting to perform circular dichroism and size exclusion chromatography experiments using the wild type hCC so that it would be possible to evaluate if a phosphatidylcholine lipid bilayer is able to induce/accelerate the process of dimerization.

Specific comments

1)     P. 1, line 4 – I believe there is an extra “and complexity”.

2)     P. 2, lines 35-36: “It may explain the processes of hCC toxicity against cells and its consequences, e.g. the occurrence of atherosclerosis or aortic aneurysm [8].” – I am not sure that the example provided by the Authors (ref 8) to illustrate how important it would be to explore the interaction of hCC with lipid membranes is the most adequate, since in that work the Authors show that the levels of cystatin C correlate inversely with the progression of the disease, whether atherosclerotic plaques or aortic dilatation.

3)     P. 2, lines 41-43: “In fact, the eukaryotic cell membranes are very heterogeneous, various types of zwitterionic alkyl phosphoholine with different alkyl chain lengths, present in both membrane leaflets constitute a substantial part of the membrane lipid content [10].” – The following work should be used concerning the membrane lipidome of an eukaryotic cell: “J.L. Sampaio et al, Membrane lipidome of an epithelial cell line, PNAS, 108 (2011) 1903-1907.”

4)     P. 2, lines 43-44: “The addition of a small amount of negatively charged SDS better simulates the vertebrae cell membrane.” – I do not fully agree with the Authors’ statement, since SDS is also not very biomimetic. I would ask the Authors to rephrase to something like “The addition of a small amount of a negatively charged phospholipid (phosphatidic acid, phosphatidylserine, phosphatidylinositol or phosphatidylglycerol) better simulates the vertebrae cell membrane.”

5)     P. 2, line 46: “microscopic techniques [12]” – I would ask the Authors to add an example of an atomic force microscopy study. Two examples are: “Hane, F. et al, Amyloid-beta Aggregation on Model Lipid Membranes: An Atomic Force Microscopy Study. J. Alzheimers Dis. 2011, 26, 485-494” and “Drolle, E. et al, Atomic force microscopy to study molecular mechanisms of amyloid fibril formation and toxicity in Alzheimer's disease. Drug Metab. Rev. 2014, 46, 207-223.”

6)     P. 2 lines 64-65: “(twenty cycles of 30 min incubation in ultrasound bath with heating and 30 min incubation in 4°C)” – Was this procedure carried out before or after the extrusion?

7)     Figure 1 and Figure 2. Besides showing the lipid concentration in each case I would also ask the Authors to make explicit the lipid:protein ratio.

8)     P. 2 line 79: “lipid phase change” should be changed to “lipid phase transition”

9)     P. 3 lines 79-80: “the temperature of the change from the gel phase to liquid phase…” please change to “the phase transition temperature from gel to liquid”

10)  P. 7, lines 152-153: “Inspection of collected data we select the residues reveals larger than average R1R2 values as residue with increase mobility in low-frequency time-frame.” – The sentence is a bit confusing and hard to understand. I would ask the Authors to rephrase it.

11)  P. 8, section 4.2. Did the Authors pass the lipid solution through the pore membrane 10 times or an odd number of times? And if the Authors were preparing 10mM stock solutions it would be better if they would pass the lipid solution through the pore membrane at least 21 times, even though the lipid solution is previously sonicated. Not that I think this can influence the overall data collected or the conclusions drawn, but this might help reducing the light-scattering problems. Besides, using such a high concentrated lipid solution and performing few passages through the pore membrane may not be sufficient to ensure that the lipid solution is homogenous (in terms of vesicle size and number of lamella) and mainly composed of unilamellar vesicles.

P. 9, section 4.4. It would be important to mention the flow rate and the detection method

Round 2

Reviewer 1 Report

The authors responded and incorporated my comments where appropriate or otherwise satisfactorily explained why this was not necessary.

Reviewer 2 Report

The explanations and clarity in the manuscript are much improved following revision and I thank the authors for taking into account my comments in a rapid manner.